# MAGIC: A tool for predicting transcription factors and cofactors driving gene sets using ENCODE data

Avtar Roopra [ID] *

Dept. of Neuroscience, 5507 WIMR, University of Wisconsin-Madison, Madison, United States of America

* asroopra@wisc.edu

## Abstract

Transcriptomic profiling is an immensely powerful hypothesis generating tool. However, accurately predicting the transcription factors (TFs) and cofactors that drive transcriptomic differences between samples is challenging. A number of algorithms draw on ChIP-seq tracks to define TFs and cofactors behind gene changes. These approaches assign TFs and cofactors to genes via a binary designation of 'target', or 'non-target' followed by Fisher Exact Tests to assess enrichment of TFs and cofactors. ENCODE archives 2314 ChIP-seq tracks of 684 TFs and cofactors assayed across a 117 human cell lines under a multitude of growth and maintenance conditions. The algorithm presented herein, **M**ining **A**lgorithm for **G**enet**I**c **C**ontrollers (MAGIC), uses ENCODE ChIP-seq data to look for statistical enrichment of TFs and cofactors in gene bodies and flanking regions in gene lists without an *a priori* binary classification of genes as targets or non-targets. When compared to other TF mining resources, MAGIC displayed favourable performance in predicting TFs and cofactors that drive gene changes in 4 settings: 1) A cell line expressing or lacking single TF, 2) Breast tumors divided along PAM50 designations 3) Whole brain samples from WT mice or mice lacking a single TF in a particular neuronal subtype 4) Single cell RNAseq analysis of neurons divided by Immediate Early Gene expression levels. In summary, MAGIC is a standalone application that produces meaningful predictions of TFs and cofactors in transcriptomic experiments.

## Author summary

Key to the control of gene expression is the level of transcript in the cell. This level is controlled large part by Transcription factors (TFs) and cofactors. TFs are DNA binding proteins that recognize specific sequence elements to control levels of gene activity. TFs recruit cofactors that do not themselves bind DNA but are brought to promoters via TFs to either enhance or repress gene expression. TFs and cofactors are thus key regulators of transcript levels. It is now routine to obtain the expression levels of every gene transcript in the genome i.e. whole transcriptome data. Understanding how the transcriptome is controlled is challenging. Herein, a method is described that predicts which Factors

**Data Availability Statement:** Expression data files are available from the GEO database (GSE84175 for CTCF WT and KO data) or TCGA (https://www.cbioportal.org/study/summary?id=brca_tcga).

**Funding:** The author received no specific funding for this work.

**Competing interests:** The author has declared that no competing interests exist.

organize and control sets of genes. The algorithm is termed **M**ining **A**lgorithm for **G**enet**I**c **C**ontrollers (MAGIC). MAGIC uses data derived from ChIPseq tracks archived at ENCODE to decipher which Factors are most likely to preferentially bind lists of genes that are altered from one biological state to another. MAGIC circumvents the principal confounds of current methods to identify Factors and will aid in the discovery of organizing principles behind large scale gene changes seen in physiology and disease.

This is a *PLOS Computational Biology* Methods paper.

## Introduction

Key to the control of gene expression is the level of transcript in the cell. This level is controlled large part by Transcription factors (TFs) and cofactors. TFs are DNA binding proteins that recognize specific sequence elements (motifs) associated with gene promoters and enhancers. TFs recruit cofactors that do not themselves bind DNA but are brought to promoters via TFs to either enhance or repress gene expression. TFs and cofactors (hereon termed 'Factors') are thus key regulators of transcript levels[1].

It is now routine to obtain the levels of every gene transcript in the genome. The datasets can contain tens of thousands of expression values per sample and the number of samples can be in the thousands such as breast cancer transcriptomes archived at The Cancer Genome Atlas (TCGA)[2]. When comparing transcriptomes from two or more conditions such as normal to cancerous tissue, thousands of mRNA levels can change. The changes reflect both alterations in tissue heterogeneity and alterations in transcriptional regulator function. We posit that in many cases, the majority of cellular transcriptome changes are driven by alterations in the function of a few Factors that coordinate gene programs. Identifying those driving Factors is a fundamental problem and therapeutic opportunity.

Current algorithms for discerning Factors behind transcriptomic changes fall into two categories. The first looks for motifs that are over-represented in the upstream regions of a set of genes[3–5]. This approach often fails because with very few exceptions, transcription factor motifs are small and often have redundant bases.Thus, searches will yield a high false positive rate; most predicted sites will not be functional. Moreover, in many genes, TFs act in concert such that the presence of a single motif element is uninformative[6]. Motif searching also precludes any prediction of cofactors because cofactors do not bind DNA directly[7]. This is especially problematic for translational research where uncovering drugable targets is a major goal, and cofactors are often enzymes that can make excellent drug targets. A second approach relies on assigning genes to Factors by analysis of ChIP-seq tracks. Genes are ranked by ChIP signal and then an arbitrary number of 'top' genes are taken as targets to form a gene set for a Factor. Fisher Exact Tests (FET) or Gene Set Enrichment Analysis-like (GSEA) analyses[8] can then be performed between differentially expressed genes and Factor sets. Examples of such algorithms are BART[9], VIPER[10], TFEA.ChIP[11], ENRICHR[12] and CHEA3[13].

ChIP-seq based approaches allow for assignment of cofactors. However, many Factors have highly non-gaussian ChIP signal distributions; the distributions can have large numbers of sites with very low but detectable signals. This thwarts unbiased attempts to define genes as 'targets' or 'non-targets'[14]. Herein, a method is described that assigns Factors to gene lists using ENCODE ChIP-seq data without first defining genes as 'targets' or 'non-targets'. The

algorithm is termed **M**ining **A**lgorithm for **G**enet**I**c **C**ontrollers (MAGIC). MAGIC circum-vents the principal confounds of current methods to identify Factors, namely: 1) the use of TF motif searches 2) inability to identify cofactors due the absence of any binding site motifs 3) assignment of Factors to genes based on hard cutoffs of ChIP-seq signals. MAGIC accepts an input set of genes and compares ChIP signals for the set with the population of ChIP signals for each Factor in ENCODE. When compared to CHEA3, TFEA.CHIP and ENRICHR, MAGIC performs favourably in predicting transcription factors and cofactors enriched in lists of differentially expressed genes.

## Methods

### MCF7 RNA profiling

MCF7 cells harboring a REST shRNA knockdown and controls were generated as described and the dataset derived from Nimblegen Arrays is described in Meyer et al[15]. Probes were considered as expressed in the experiment if they were called as 'present' in all 3 samples of either controls or knockdown cells. If a probe was present in only one condition, the values in the other condition were accepted regardless of the present/absent call. Student t tests were performed for each probe and one probe was assigned per gene symbol by taking the probe with the largest T statistic between conditions. Genes up-regulated more than 3 Standard Deviations from the log of the mean fold change (average shREST value/average shCon value) and having a Benjamini Hochberg[16] corrected t-test p value<0.05 were used for Factor anal-ysis. The background list consisted of all symbols found to be present in all controls or all knockdowns or both (S1 Table).

### TCGA profiling

Breast cancer RNAseq data (data_RNA_Seq_v2_expression_median.txt) was downloaded from cBioPortal (latest datafreeze as of 6July2017). PAM50[17] designations were used to parse tumors into Luminal-A or Basal subtypes. A gene with an RPKM $\geq 1$ in at least 50% of either basal or Luminal-A samples was considered for further analysis. Fold changes, statistics and gene lists were then generated as above (S2 Table).

### Sams et al profiling

RNAseq data associated with Sams et al[18] (GSE84175) was downloaded from GEO. A gene with an RPKM $\geq 1$ in all controls or all CTCF KO cells was considered for further analysis and fold changes, statistics and gene lists were then generated as above for REST knock-down cells. 17,015 genes were expressed by this criterion with 432 genes down-regulated at least 1.5 fold with a corrected t-test p value $< 0.25$ (S3 Table).

### Jaeger et al profiling

Supplemental file 41467_2018_5418_MOESM6_ESM.xlsx from Jaeger et al [19] was used to generate gene lists for Factor analysis. All genes with a positive test statistic and adjusted p-value<0.05 (842 genes) were accepted as up-regulated genes from the DG_Reactive_HC tab. All 5306 detected genes were used as a background list (S4 Table).

### MAGIC

MAGIC aims to determine whether genes in a query list are associated with higher ChIP values than expected by chance for a given transcription factor or cofactor based on ChIP-seq tracks

archived at ENCODE. MAGIC was written in Python 3.7 and packaged into a stand-alone executable using Pyinstaller 3.6.

## Generation of MAGIC matrices

MAGIC utilizes MAGIC matrix files (*.mtx files) to determine Factors behind gene lists. MAGIC matrices were generated as follows: All human genes were assigned a gene domain encompassing one of the following: 1) gene body (promoter to the end of the last exon) plus 1Kb flanking sequence either side of the gene body, 2) gene body plus 5Kb flanking sequence either side of the gene body, 3) 1Kb upstream of the promoter and 300bp downstream. NarrowPeak files for all Factors were downloaded from: https://www.encodeproject.org/matrix/? searchTerm=chip-seq&type=Experiment&target.investigated_as=transcription+factor&files. file_type=bed+narrowPeak&award.project=ENCODE

A custom Python script was used to extract the highest ChIP-seq peak value (signalValue) within the gene domain for each ChIP-seq track. Thus each Factor and gene were assigned a single signalValue. Matrices were generated that either include Chip-Seq tracks for Factors that were engineered (for example by CRISPR or stable transfection) in which case they have "_with_engineered_factors" in the name, or do not include engineered factors. Five matrices are therefore currently available to the user:

1Kb_gene.mtx
1Kb_gene_with_engineered_factors.mtx
1Kb_promoter.mtx
1Kb_promoter_with_engineered_factors.mtx
5Kb_gene.mtx
5Kb_gene_with_engineered_factors.mtx

Matrices with engineered factors are derived from 2312 ChIP-seq tracks covering 684 Factors in 588 cell lines or conditions. Matrices without engineered Factors derive from 1923 ChIP-seq tracks covering 392 Factors in 207 cell lines or conditions. All Matrices encompass 27,941 genes.

## Defining enriched factors in a gene set

At least two lists are entered into MAGIC. The first is a background list containing all genes of relevance to the experiment (for example, all genes present or expressed in an experiment). Subsequent lists contains genes of interest–the query lists (e.g. differentially expressed genes). Query lists are subsets of the background list and both lists are filtered by MAGIC for genes present in the designated MAGIC matrix.

For each Factor in the Matrix, MAGIC orders the N genes in the accepted background list by ChIP signal (lowest to highest). It then generates a normalized cumulative distribution function (CDF) such that:

$$B(c) = \frac{1}{N} \sum_{i=1}^{N} 1_{B_i \leq c}$$

Where B(c) is the fraction of genes in the background list (N genes) with ChIP signal less than or equal to c. B are the ordered ChIP signals. $1_{B_i \leq c}$ is an indicator function that is equal to 1 if $B_i \leq c$ else equal to 0. An empirical distribution function, Q(c) is generated by accumulating the ordered X query genes (Q) (where Q is a subset of B and $1_{Q_i \leq c} = 1$ if $Q_i$ less than or

equal to c else equal to 0):

$$Q(c) = \frac{1}{X} \sum_{i=1}^{X} 1_{Q_i \leq c}$$

The maximal difference between the two cumulatives is then calculated thus:

$$D_{sup} = sup_c(B(c) - Q(c))$$

A Kolmogorov-Smirnov test is performed on N, X and $D_{sup}$ to determine the probabilty of this difference or larger occuring by chance. The argument of $D_{sup}$ ($arg_{Dsup}$) is also noted; this is used to define that subset of the query list that is targeted by the Factor. If there are fewer than five genes with chip values greater than $arg_{Dsup}$, the Factor is triaged and not considered further.

A Score for each Factor is calculated to incorporate the Benjamini-Hochberg corrected Kolmogorov-Smirnov p value, as well as a measure of how the highest ChIP values in the query list compare to the highest values in the background list. Thus, for each Factor, the mean of the top n chip signals in the query list of length X, is compared to the mean of the top n ChIPs in the population to produce a ratio such that:

$$n = 0.05X$$

and

$$r = \frac{\mu_q}{\mu_b}$$

Where $\mu_q$ is the mean for the top n signals for Q and $\mu_b$ is the mean of the top n signals for B.

A Score, S, is assigned to each Factor thus:

$$S = -\log(P_{corr}) \times r$$

Where $P_{corr}$ is the Benjamini-Hochberg corrected Kolmogorov-Smirnov p value. Factors are then sorted by Score.

**Implementation of MAGIC.** A tab delimited text file is requested by MAGIC (lists file). The first column contains a list of all genes expressed in the experiment (background list). Any number of other columns are then added containing query lists. For example, a query list may be all genes that go up under some criterion and another may be all genes that go down. The first row is the header and must have unique names for each column. MAGIC then requests which Matrix to use.

MAGIC analyzes each query list and generates a series of output files and sub-directories in the directory containing the original lists file. An 'Accepted_Lists.txt' file is generated which is the original input file filtered for genes in the Matrix. A Platform_Matrix.txt file is the matrix file filtered for genes in the the background list. A series of directories are generated named after each query list in the lists file. Each directory contains sub-directories and files for the stand-alone analysis of that query list. A Query_List_Details.xls file contains statistical information for all non-triaged Factors. All Factors associated with $P_{corr} < 10\%$ are highlighted in bold red. Data reported in Query_List_Details.xls are:

Factor Name of Factor

Description The ENCODE cell line, tissue or experiment description

Critical ChIP: $\arg_{Dsup}$ i.e. ChIP value at $D_{sup}$ (This value is used to determine target genes in the list)

Obs Tail Mean: Average of the 95th percentile ChIP values (n values) in the query list.

Exp Tail Mean: Average of the top n ChIP values in the background.

Tail Enrichment: Ratio of the Obs and Exp Tail Means (r)

Raw P: Kolmogorov-Smirnov p value

Corrected P: Benjamini Hochberg corrected p value ($P_{corr}$)

Score: Score ($-\log(P_{corr}) \times r$)

Where ENCODE contains multiple Chip-seq tracks for a Factor, the best scoring is also reported in a Summary.xls file with the same layout as above–each Factor appears once in this file. A query_list_summary.pdf contains a bar graph of Factors and Scores with $P_{corr} < 10\%$.

Query_list_Drivers.gmx is a tab delimited file in the GMX format utilized by GSEA[8]. The first column contains the background list of genes. The second contains the query list. Subsequent columns contain all target genes of each Factor. Target genes are defined as those genes in the query list whose ChIP signal is greater than the Critical ChIP ($\arg_{Dsup}$) i.e. the ChIP at which there is the maximal difference between the population and query cumulative.

A sub-directory called 'CDFs' contains graphical displays of the analysis for all non-triaged Factors. The naming format is 'rank (*integer*)'_'factor (*string*)'.pdf (e.g. 1_NRSF.pdf; rank = 1, factor = NRSF) (S1 Fig) where ranking is determined by Score. Two cumulative functions are displayed: the black curve is the fractional cumulative of all genes in the background list against ChIP values, red is the same for query genes. A blue vertical line denotes the ChIP value at $D_{sup}$ i.e. $\arg_{Dsup}$. Red ticks along the x-axis represent each gene in the query list and black ticks are all genes in the background. Red ticks with circles ('lollipops') are the n = 0.05X best chiped genes. Black lollipops are genes in the background list with the n highest ChIP values.

A second sub-directory called 'Auxiliary_Files' is populated with data behind the summary files. The 'query_list_raw_results.CSV' file contains the same columns as the Query_list_Summary.xls file but has raw data for all factors including those that were triaged and not considered for further statistical analysis. It also contains the Kolmogorov-Smirnov D statistic for each factor. D statistics with a negative sign denote D values for triaged factors; the negative sign is used by the algorithm for triage sorting.

Query_list_Sub_Matrix.txt is the MAGIC Matrix filtered for genes in the query list.

Triaged_Factors.txt is a list of factors that were not considered (triaged).

Triaged_Genes.txt contains all genes in the query that were not in the MAGIC Matrix and therefore eliminated from analysis.

A sub-directory named 'Target_Data' contains comma separated text files for each Factor with $P_{corr} < 10\%$. Each file contains a list of target genes for that Factor and associated MAGIC Matrix ChIP value.

## Obtaining MAGIC

MAGIC can be downloaded from: **https://go.wisc.edu/magic**

The Python script and information on how to access required Matrix files is available at: https://github.com/asroopra/MAGIC

## Defining cofactors and associated TFs of manipulated factors for ROC, PR and rank analysis

To generate a list of potential cofactors and transcription factors that may interact physically or functionally with a manipulated Factor (hereon called Potential Interactors, PIs), the file

9606.protein.links.detailed.v11.0.txt was downloaded from the STRING protein interaction database (http://string-db.org/)[20]. Proteins with a Combined Score of greater than 300 (representing the 75th percentile of scores) were taken as PIs.

## Precision Recall (PR) and Receiver Operator Characteristics (ROC) analysis

Outputs from MAGIC, CHEA3, TFEA and Enrichr were ordered by Score (for MAGIC) or Fisher's Exact Test p value and ranks were scaled between 1/n and 1 for each algorithm and library as in Keenan et al[13]. This produced vectors of positive and negative calls for each test (Calls Vectors). The positive class was defined as the manipulated Factor and all PIs as defined above. ROC and PR curves and statistics were generated by importing roc_curve and precision_recall_curve from sklearn.metrics in Python3.7. No balancing of imbalanced datasets was performed and so PR analysis was prioritized over ROC analysis.

## Rank analysis

To determine how well the algorithms prioritized the manipulated Factor and PIs, scaled ranks of the positive calls in the Calls Vectors were cumulated to give D(r). A lack of enrichment of TFs and associated cofactors for either high or low ranks would produce a uniform distribution (which equates to r). Thus D(r)-r is plotted for each dataset: the greater the Area Under the Curve (AUC), the better the identification of TFs and cofactors. Contiguous runs of positive calls (a block of positive calls) are assigned the rank of the first element in the block. Differences between the distributions were assessed using Kolmogorov-Smirnov tests.

## Results

The purpose of MAGIC is to identify those transcription factors and cofactors responsible for patterns of gene expression changes after a manipulation or between conditions. For all analyses, the 1Kb_gene.mtx matrix was used. MAGIC was tested on four transcriptome datasets: **1) MCF7(shCon_vs_shREST)**. This dataset consists of MCF7 breast cancer cells stably expressing shRNA against the transcription factor REST or a control sham shRNA. REST is a transcriptional repressor and tumor suppressor whose loss drives tumor growth of MCF7 cells in mouse xenograft model of breast cancer[15, 21, 22]. This simple system consists of a clonal cell line lacking a single factor. 119 genes out of 15,446 expressed genes were identified as up-regulated by more than 3 standard deviations than the log mean fold change and an associated False Discovery Rate (FDR)<5% (S1 Table). **2) TCGA(Lum_vs_Basal)**. 233 Luminal-A and 79 Basal-like tumors were identified in The Cancer Genome Atlas (TCGA) Nature 2012 provisional breast cancer dataset [23]. Luminal-A tumors are a subset of breast cancers defined by their robust expression of estrogen receptor alpha (ERα) and associated pioneer factors. They also express the progesterone and her2 receptors. Basal-like tumors are estrogen, progesterone and her2 receptor negative. This is a complex dataset with over 300 samples and heterogeneous tissue. 203 genes out of 17,814 expressed genes were up-regulated in Lum-A over Basal tumors using the same criterion as for MCF7(shCon_vs_shREST) (S2 Table). **3) Brain (WT_vs_CTCFko)**. MAGIC utilizes ChIP-seq data derived from immortalized or transformed human cancer cell lines and the above two examples utilize either an immortalized cell line or cancer tissue. To test whether MAGIC could be used on data derived from non-cancerous, highly heterogeneous, rodent samples, we turned to non-malignant mouse brain tissue. CTCF is a TF that organizes long distance interactions in the genome[24]. Sams *et al* demonstrated that elimination of CTCF in excitatory neurons (while sparing its expression in other neuron subtypes, astocytes and glia) results in defects in learning, memory and neuronal plasticity[18].

432 genes were down-regulated upon CTCF loss out of 17,015 expressed genes (S3 Table). 4) *DGC(Quiet_vs_Reactive)*. Single cell transcriptome profiling allows the discerning of responses of individual cells to stimuli. Jaeger *et al* performed single cell profiling on mouse dentate granule cells in response to a novel location stimulus [19]. Cells were sorted on Fos expression, an Immediate Early Gene (IEG) that is expressed in response to neuronal firing [25]. 843 genes were up-regulated in Reactive Dentate Granule cells out of 5305 expressed genes (S4 Table).

## Identification of manipulated factors

Arguably, the most impactful output of any Factor discovery tool is the best ranked, or 'top' Factor predicted to target a list of input genes. The above datasets were subjected to Factor analysis using MAGIC, TFEA.ChIP[11], Enrichr[12] and CHEA3 [13]. Integer ranks were calculated for the manipulated Factor in each experiment ('top' Factor has rank = 1). Reciprocals were calculated so that unity represents the best rank and zero the poorest. Fig 1 and S5–S8 Tables shows that MAGIC was able to call the manipulated Factor in each experiment as the top ranked Factor for 3 of the datasets i.e. REST in MCF7(shCon_vs_shREST) (S5 Table), ESR1 in TCGA(Lum_vs_Basal) (S5 Table), and CTCF in Brain(WT_vs_CTCFko) (S7 Table). MAGIC called FOS as 2nd rank in the fourth dataset, DGC(Quiet_vs_Reactive) (S8 Table), having called FOSL2 as top rank. Thus MAGIC was able to identify the manipulated Factor as well as, or better than CHEA3 using any of its libraries, TFEA or Enrichr.

## Identification of cofactors and Potential Interacting TFs

Transcription factors work by recruiting cofactors via protein-protein interactions and often work in unison with other TFs[26]. Thus it would be useful if an algorithm designed to identify TFs behind gene lists could also highlight associated cofactors and TFs (i.e. Potential Interactors—PIs). Ideally those cofactors would appear near the top of the list of putative Factors. To assess how well MAGIC compares to the other algorithms in identifying PIs associated with DEG lists, a cohort of PIs was identified for the above 4 manipulated TFs. Annotated protein-protein interactions were downloaded from the STRING database (https://string-db.org)[20] and used to identify proteins associated with either REST, ESR1, CTCF or FOS with a STRING Combined Score of greater than 300 (representing the 75th percentile of scores) [27]. Fractional ranks for the manipulated Factor and PIs were then cumulated (to give D(r)). A lack of enrichment of TFs and associated cofactors for either high or low ranks would produce a uniform p-value distribution (which equates to r). Thus D(r)-r was plotted for each dataset: the greater the Area Under the Curve (AUC), the better the identification of TFs and cofactors. This approach is borrowed and modified from Keenan *et al* in their excellent publication describing CHEA3[13]. Fig 2A and 2B shows that MAGIC manifests the largest AUC for REST and its PIs in MCF7(shCon_vs_shREST), ESR1 and associated PIs in TCGA(Lum_vs_-Basal), CTCF and PIs in Brain(WT_vs_CTCFko) and for FOS in DGC(Quiet_vs_Reactive). In an attempt to integrate the rank results for the manipulated Factor in Fig 1 and recollection of PIs in Fig 2, the AUC for D(r)-r functions was multiplied by the fractional rank values of the manipulated Factors in order to weight the AUCs according to priority the algorithms placed on the manipulated Factor. Using this 'weighted AUC' metric, MAGIC manifested a robust ability to call the manipulated Factor along with its PIs (Fig 2C and 2D)

## Precision Recall and Receiver Operator Characteristics

Using the manipulated Factors (REST, ESR1, CTCF and FOS) and their PIs as defined above, Precision Recall (PR) and Receiver Operator Characteristics (ROC) curves were generated for

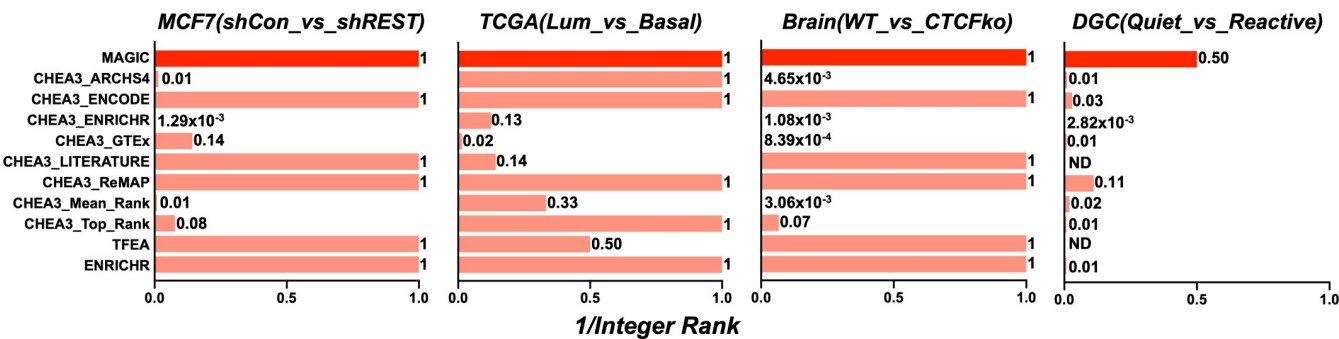

**Fig 1. Comparing ranks of manipulated factors predicted by MAGIC, CHEA3, TFEA and Enrichr.** MCF7(shCon_vs_shREST), TCGA(Lum_vs_Basal), Brain(WT_vs_CTCFko) and DGC(Quiet_vs_Reactive) datasets were analyzed by MAGIC, CHEA3 (using all available libraries: ARCHS4 co-expression, ENCODE ChIP-Seq, Enrichr Queries, GTEx co-expression, Literature mining, ReMAP ChIP-seq, Mean Rank and Top Rank), TFEA and Enrichr. The reciprocal integer ranks for REST, ESR1, CTCF and FOS (the Factors manipulated in MCF7(shCon_vs_shREST), TCGA(Lum_vs_Basal), Brain(WT_vs_CTCFko) and DGC(Quiet_vs_Reactive)) are plotted. The top rank = 1, second rank = 0.5 etc. ND = Not Determined; Factor not present in library.

the 4 datasets, algorithms and libraries to assess how well MAGIC compares in classifying Factors behind input gene lists. The datasets are unbalanced in that the positive class (consisting of the manipulated Factor and PIs) is smaller than the negative class. Therefore the PR curve was prioritized in assessing algorithm performance. Fig 3A–3C show that MAGIC manifests superior precision for the MCF7(shCon_vs_shREST), TCGA(Lum_vs_Basal), and Brain (WT_vs_CTCFko) datasets with PR AUCs of 84%, 83% and 91% respectively. For the DGC (Quiet_vs_Reactive) dataset, MAGIC exhibited an AUC = 72%, 2nd to TFEA at AUC = 77%. When normalizing the PR AUC for ranking of the manipulated Factor as in Fig 2C, MAGIC exhibited superior performance across all 4 datasets (Fig 3D). For completeness, ROC curves were generated for all algorithms across the four data sets (Fig 3B). ROC analysis showed that MAGIC performed solidly across all four datasets and was bested by only one algorithm in one dataset (CHEA3_Mean_Rank in DGC(Quiet_vs_Reactive)).

## The Requirement for a valid background gene list

To test how MAGIC might be sensitive to the background list of genes, MAGIC was tested on the 4 datasets in the absence and presence of a bespoke background list. The background lists for MCF7(shCon_vs_shREST), Brain(WT_vs_CTCFko) and DGC(Quiet_vs_Reactive) were generated by accepting all genes in the dataset that had detectable signal in all samples of either control or experimental condition. For TCGA(Lum_vs_Basal) which has 312 samples, this was modified to allow genes that were detectable in at least 50% of samples in either condition. The results using these background lists were compared to results obtained using all genes in the matrix (i.e. 'minus background'). MAGIC was first assessed for its ability to call manipulated Factors and PIs as in Fig 2 using D(r)-r plots. Fig 4A shows that the D(r)-r AUC is significantly reduced in the absence of a background list for all four datasets indicating a reduced ability to call Factors. Precision was also reduced in the absence of a background list as judged by PR AUC and precision at 80% recall (Fig 4B). ROC parameters were also diminished in the absence of a background list. Thus ROC AUC as well as the True Positive Rate measured at 20% False Discovery Rate was reduced in the absence of a background list (Fig 4C).

To test the effect of background list on p values associated with Factors and PIs in MAGIC outputs, emperical cumulatives were generated of *-log(FDR)* values (Benjamini Hochberg corrected p values) for the four datasets. Fig 4D shows that there are significantly more low FDR calls in the absence of a background list demonstrating that lack of a background list inflates statistical significances associated with MAGIC outputs. Finally, the effect of background list

**A**

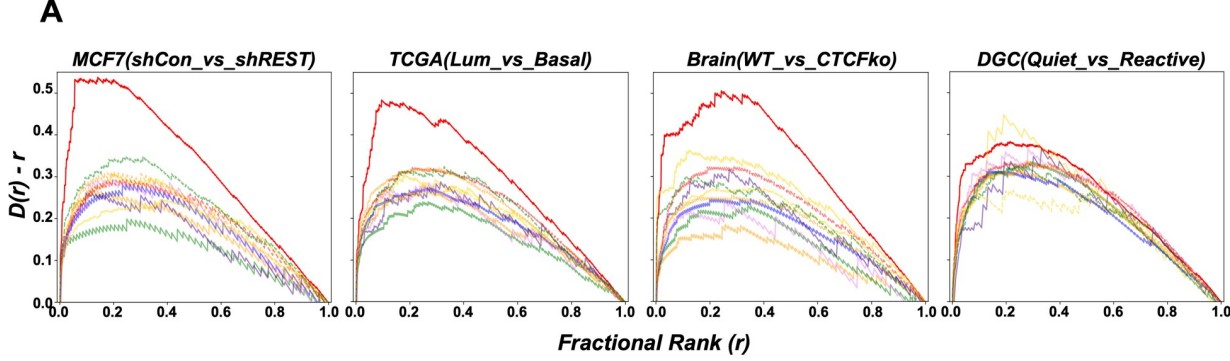

**B**

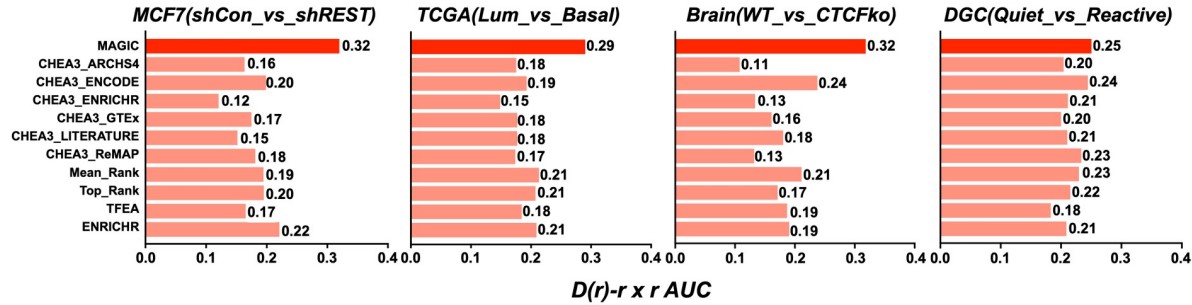

**C**

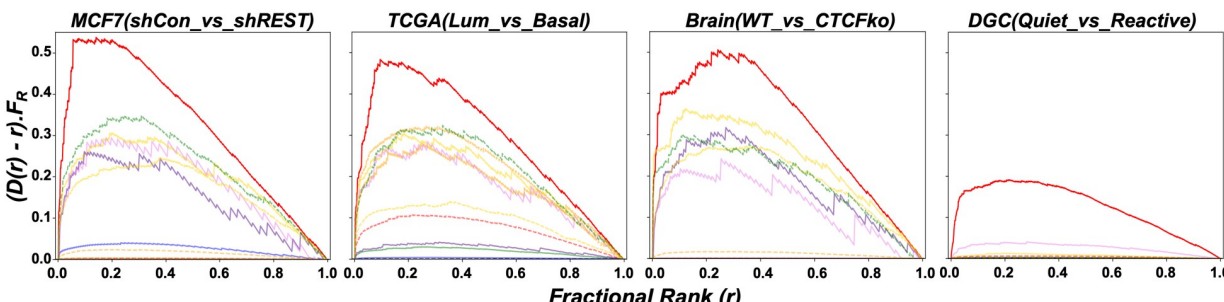

**D**

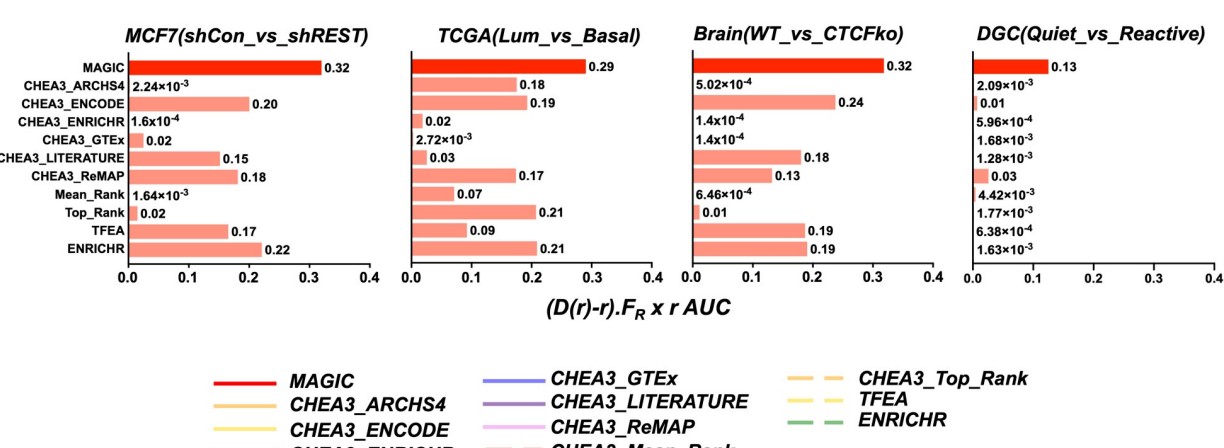

**Fig 2. Manipulated transcription factors and associated cofactors are preferentially ranked by MAGIC compared to CHEA3, TFEA and Enrichr.** (A) Emperical cumulatives were generated of factional ranks (1/Integer ranks) for manipulated factors and associated cofactors for the 4 datasets using all algorithms and libraries as in Fig 1. The difference between the cumulative of all scaled fractional ranks and a uniform distribution for the manipulated factor and associated cofactors is plotted against the fractional rank. Kolmogorov-smirnov tests of each distribution against a uniform distribution yields p<$10^{-10}$ for all tests. (B) Area Under Curve (AUC) for D(r)-r x r curves in panel A. (C) The D(r)-r curves in panel A were scaled for the rank of the manipulated actor. For each algorithm, D(r)-r was multiplied by the fractional rank of the manipulated factor ($F_R$). (D) AUCs for curves in panel C.

was tested on ranking of Factors. The integer ranks for the 50 top ranked Factors for each dataset in the presence of appropriate background lists was compared to their ranks in the absence of a background list. Fig 4E shows that Factor order is impacted by the absence or presence of the background.

In summary, using four disparate datasets, MAGIC performed at least as well as CHEA3, TFEA and Enrichr according to ranking of the manipulated Factor, calling of potential interactors, PR characteristics and ROC analysis.

## Discussion

MAGIC is a standalone application for predicting transcription factors and cofactors that drive gene expression changes in transcriptomic experiments. Data presented herein shows that MAGIC compares favourably with three other algorithms: CHEA3, TFEA and Enrichr. A number of approaches have been used historically to tackle the difficult problem of determining which Factors drive sets of genes. One approach—exemplified by oPossum[5]—relies on searching for short transcription factor binding motifs in promoter proximal sequences. Using hypergeometric tests provides a likelihood of enrichment for the factor. This approach is limited due to the small size of binding sites for most factors; a random 6bp sequence will be found approximately 750,000 times in the human genome. This approach also cannot predict cofactors as cofactors do not bind DNA directly.

A more sophisticated approach relies on comparing query lists of genes with gene lists attributed to Factors based on ChIP-seq experiments. This approach is the basis for CHEA3, TFEA and Enrichr. ChIP-seq based allocation of genes to Factors ensures that only actual binding events are considered but even here, genes can be falsely attributed to Factors because many sites where Factors are ChIPed are not functional[3]. Thus many genes will show some low level of binding/chip signal. For example, extracting all REST ChIP-seq tracks from wgEncodeRegTfbsClusteredV2.bed from the UCSC genome browser (http://hgdownload.soe.ucsc.edu/goldenPath/hg19/encodeDCC/wgEncodeRegTfbsClustered/) showed 27,386 binding sites across the human genome with 17,971 genes showing a detectable signal within 5Kb of the promoter. This thwarts unbiased attempts to define genes as 'targets' or 'non-targets'. These caveats not withstanding, ChIP-seq based allocation of genes to Factors and Fisher Exact Tests give good performance in real world tests[9, 11–13]. MAGIC attempts to overcome the problem of assigning genes to Factors by utilizing complete ChIP-seq profiles without parsing Factors into 'bound' and 'not bound' classes.

CHEA3, TFEA (using the 'Association analysis' option) and Enrichr use Fisher Exact Tests (FET) to assess associations between input gene lists and Factor targets[11–13]. The rank order of Factors driving gene lists is therefore not altered by the size of the universal set of all genes (the background gene list) in the analysis; FET p values scale with background gene list size. In contrast, MAGIC output is sensitive to the background gene list. For each Factor in the chosen matrix, MAGIC performs a Kolmogorov-Smirnov test on cumulatives of all ChIP-seq signals in the background list of genes and the input list. The Score combines the corrected p value from this test and a measure of how skewed the chip signals are towards high values in

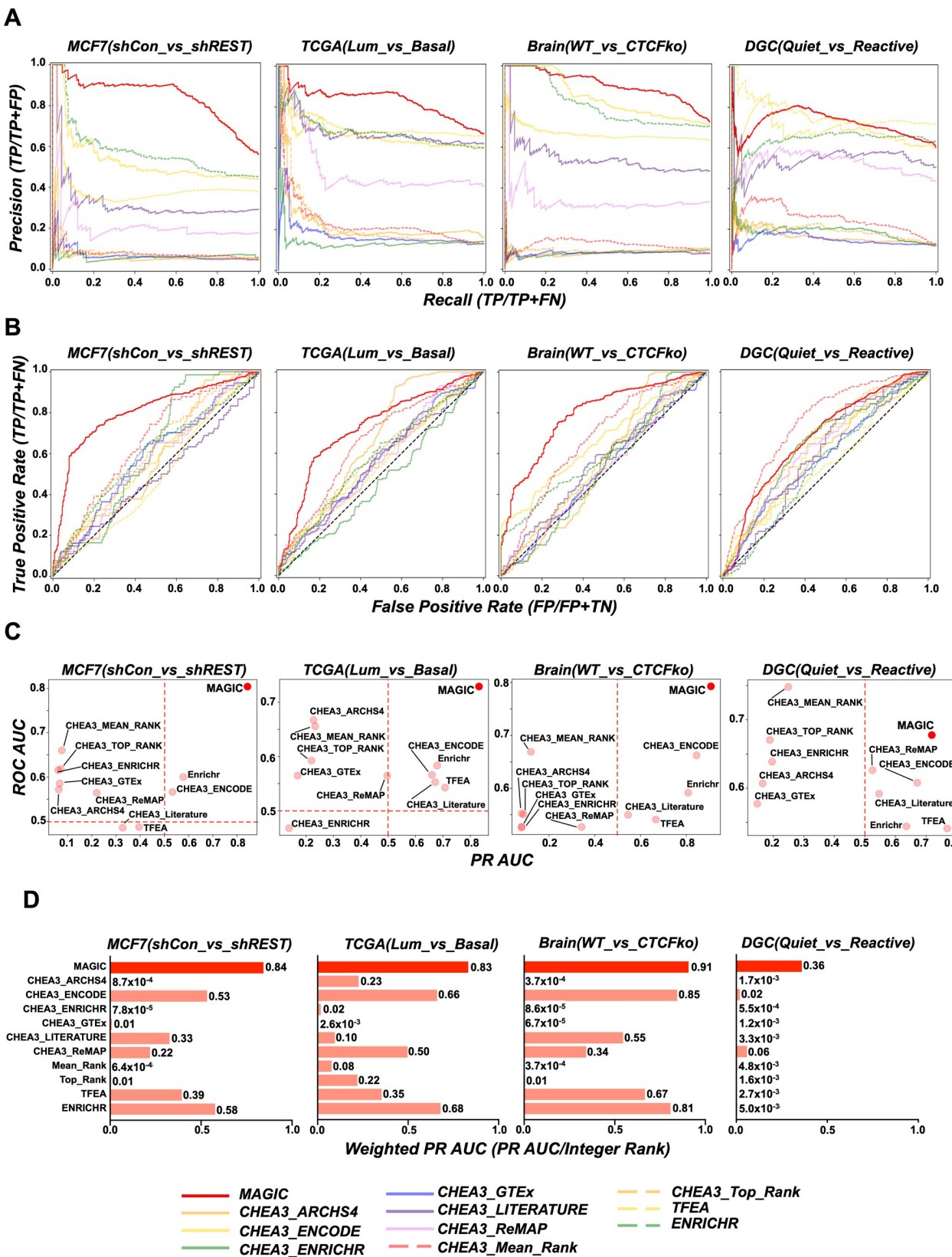

**Fig 3. MAGIC demonstrates skill at calling manipulated factors as assessed by Precision Recall and Receiver Operator Characteristics.** (A) Precision Recall curves for the four datasets and all algorithms and libraries. (B) Receiver Operator Characteristic curves for the four datasets and all algorithms and libraries. As in panel A, data was not balanced prior to graphing. (C) ROC versus PR AUCs for all algorithms and libraries. (D) PR AUCs were scaled for fractional rank of the manipulated factor by multiplying PR UAC by $F_R$.

the query versus background lists (see Methods). It follows then, that the output (p values, Scores and ranks) are sensitive to the background lists (Fig 4). Unlike FET p values that scale with background list sizes and do not change rank order, MAGIC p values have a complicated relationship with background gene content. The shape of the background cumulative will change depending on the set of genes in it, and so the maximal distance between it and the input cumulative (i.e. the Kolmogorov-Smirnov D statistic) will either increase or decrease depending on the argument of D.

Testing MAGIC on cells transfected with REST shRNA highlights the ability of the algorithm to not only identify transcription factors but also their associated cofactors. MAGIC called REST as the highest scoring factor but also called SIN3A, HDAC2, CoREST, G9a and CtBP2, which are well-characterized REST cofactors[28–31]. EZH2 and SUZ12 were also called as factors that preferentially bind up-regulated genes. Though there are reports of REST interacting with Polycomb[32], it is likely that Polycomb targets many of the same genes as REST but does so independently[33]. Interestingly the second highest scoring Factor was SP2 that has no reported association with REST (S9 Table). Current work in the lab is focussed on testing the hypothesis that REST and SP2 do interact–perhaps in a similar manner to REST and SP3[34].

Using gene lists derived from over 300 breast tumor transcriptomes (S10 Table), MAGIC correctly called ERα (encoded by the ESR1 gene) as the principal TF in ER positive (Luminal-A) versus negative (Basal-like) tumors. ERα binds chromatin with pioneer factors and MAGIC sucessfully called GATA3, FOXA1 and FOXA2[35]. P300 and EZH2 are ERα cofactors and were called by MAGIC[36, 37].

ENCODE hosts ChIP-seq tracks derived from transformed or immortal cell lines and human embryonic stem cells. A particular concern was that generating a matrix from homogenous cell lines may preclude MAGIC's use in analysis of transcriptome data derived from in vivo samples that are not immortalized and likely heterogeneous. However, testing of MAGIC on CTCF knock-out mouse brains demonstrates that gene lists derived from control and experimental whole brain extract are handled appropriately (S11 Table). In GSE84175, CTCF was deleted in a subset of post-mitotic cells (eliminated from excitatory neurons but retained in all other neuronal subtypes)[18, 38]. MAGIC correctly called CTCF as the principle factor in the experiment. The CTCF binding partner RAD21 was also called and would reflect its role as part of the cohesin complex[39, 40]. Interestingly, ZNF143 was also in the top 5 hits. This is in keeping with the recent findings of Mourad and Cuvier[38] showing that CTCF and ZNF143 coordinate chromatin border domain formation. Thus, MAGIC was able to call CTCF as well as a known associated proteins when provided transcriptome data from a highly heterogeneous tissue where only a small subset of cells were altered in the experiment.

Single cell RNA-seq on hippocampal dentate granule cells from mice subjected to novel environments shows differences in transcriptomes between sparsley firing neurons in the dentate gyrus versus quiet cells[19]. When profiles are sorted based on FOS expression, MAGIC was able to call FOS as the second most likely driver of gene changes, placing the highly homologous protein FOSL2 as top rank[41]. This prediction was superior to any of the other algorithms.

All analyses in this manuscript utilized the 1Kb_gene.mtx matrix. Nearly identical results were obtained with the 5Kb_gene.mtx. Future work will focus on the relative performances of

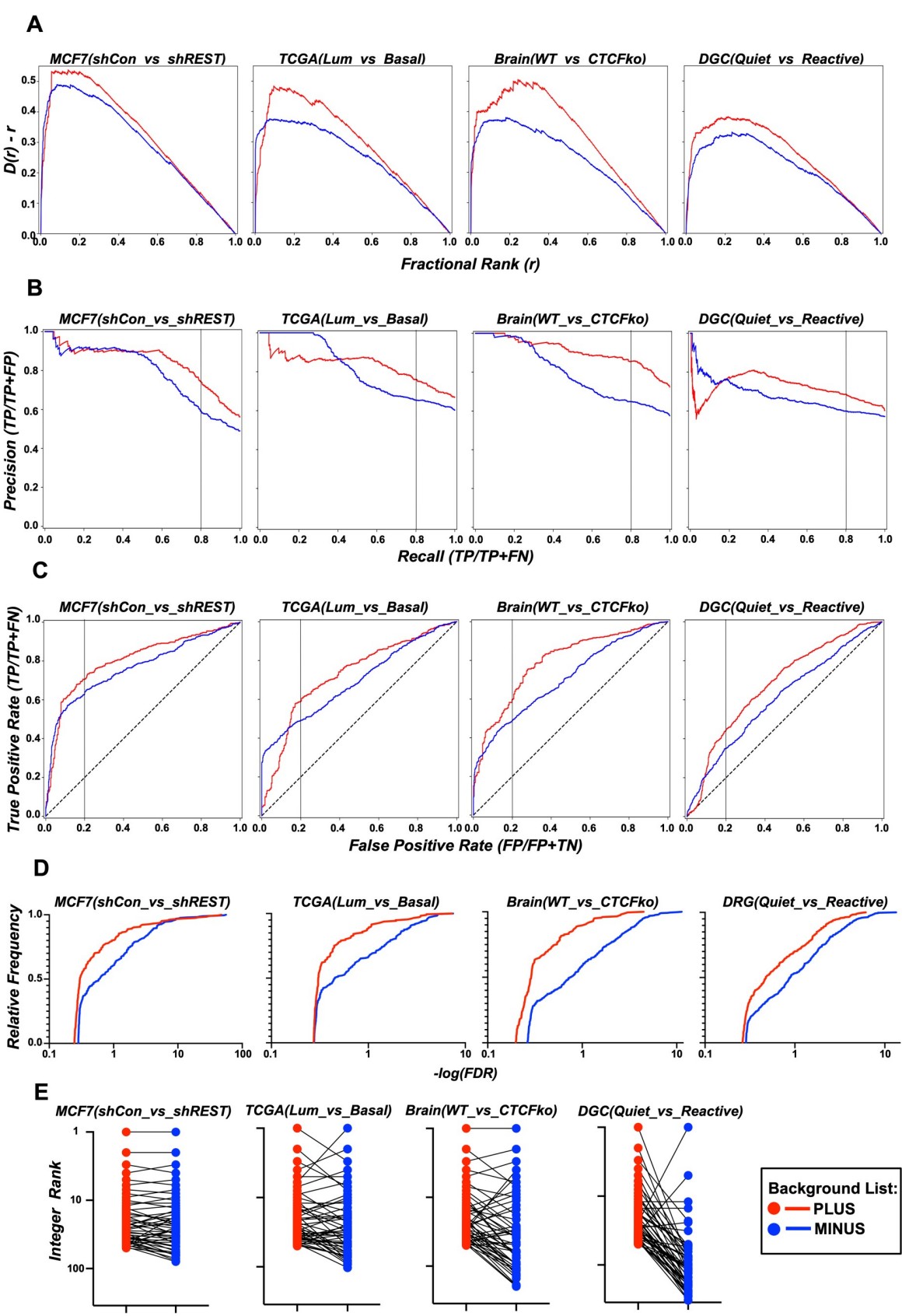

**Fig 4. MAGIC requires a valid background gene list for optimal performance.** (A) D(r)-r curves for the 4 datasets generated for MAGIC outputs in the presence or absence of a background list. Kolmogorov-smirnov statistics for the 2 curves: MCF7 (shCon_vs_shREST); D = 0.10, p = 8.6x10$^{-5}$. TCGA(Lum_vs_Basal); D = 0.15, p = 4.6x10$^{-11}$. Brain(WT_vs_CTCFko); D = 0.26, p = 1.1x10$^{-16}$. DGC(Quiet_vs_Reactive); D = 0.11, p = 3.9x10$^{-9}$. (B) Precision Recall curves for MAGIC outputs in the presence or absence of a background list: MCF7(shCon_vs_shREST); 0.84 vs 0.78, TCGA(Lum_vs_Basal); 0.83 vs 0.81, Brain(WT_vs_CTCFko); 0.91 vs 0.71, DGC(Quiet_vs_Reactive); 0.72 vs 0.67. Black vertical line denotes 80% Recall (C) Receiver Operator Characteristics curves for MAGIC outputs in the presence or absence of a background list (unbalanced). (D) Emperical cumulative distribution for False Discovery Rates associated with MAGIC outputs in the presence or absence of a background list. For all datasets, Kolmogorov-smirnov p $< 10^{-4}$. (E) The integer ranks for the top 50 Factors called by MAGIC in the presence of a background list were compared to their ranks in the absence of a background list.

the matrices provided. It should be noted that none of the matrices incorporate information from distal enhancers which are clearly key to fully understanding gene regulation[42]. Current efforts are underway to incorporate long distance information into the MAGIC algorithm. A prototype of the MAGIC approach was recently used to identify EZH2 as a principle driver of gene changes in epilepsy[43]. The approach was validated by biochemical analysis of mouse brains that showed an almost order of magnitude increase in levels of this protein over a 10 day window after seizure induction and robust repression of target genes. This prototypical approach aggregated all ChIP-seq tracks for a Factor into one meta-track and so lost all information on cell type and experimental conditions. Also, a Mann-Whitney test was incorporated instead of the more sensitive Kolmogorov-smirnov test. Nevertheless, it provided an extremely powerful prediction of EZH2 induction. MAGIC concurs with the prototypical algorithm.

MAGIC was tested against CHEA3, TFEA and Enrichr on 4 datasets where it performed well. However, the general performance level of MAGIC against the leading algorithms will only be truly revealed with significantly increased numbers of tests. In conclusion, MAGIC was able to call principle transcription factors and cofactors that target gene lists with a performance comparable to or better than current leading brand algorithms.

## Supporting information

**S1 Fig. Example of a CDF graphic produced by MAGIC.** Two cumulative functions are displayed: the black curve is the fractional cumulative of all genes in the background list against ChIP values, red is the same for query genes. A blue vertical line denotes the ChIP value at D$_{sup}$ i.e. arg$_{Dsup}$. Red ticks along the x-axis represent each gene in the query list and black ticks are all genes in the background. Red ticks with circles ('lollipops') are the n (n = 0.05X) best chiped genes. Black lollipops are genes in the background list with the n highest ChIP values. (TIFF)

**S1 Table. Background and input list for MCF7(shCon_vs_shREST).** The first column contains 15445 genes in the background list. Second column contains 118 genes induced as described in Methods and Results. (TXT)

**S2 Table. Background and input list for TCGA(Lum_vs_Basal).** The first column contains 17815 genes in the background list. Second column contains 203 genes induced as described in Methods and Results. (TXT)

**S3 Table. Background and input list for Brain(WT_vs_CTCFko).** The first column contains 15509 genes in the background list. Second column contains 203 genes down-regulated as

described in Methods and Results.
(TXT)

**S4 Table. Background and input list for DGC(Quiet_vs_Reactive).** The first column contains 5305 genes in the background list. Second column contains 842 genes induced as described in Methods and Results.
(TXT)

**S5 Table. REST interactors and rank-ordered algorithm outputs for MCF7(shCon_vs_shREST).** The first column contains proteins associated with REST extracted from the STRING database with a Combined Score>300. Subsequent columns contain ordered lists of Factors predicted to target the input list by the varous algorithms and associated libraries. If the same Factor appears multiple times, it is because there are multiple entries (e.g. ChIP-seq tracks in different cell lines) for that factor in the libraries mined by the algorithm.
(TXT)

**S6 Table. Estrogen Receptor alpha (ESR1) interactors and rank-ordered algorithm outputs for TCGA(Lum_vs_Basal).** The first column contains proteins associated with ESR1 extracted from the STRING database with a Combined Score>300. Subsequent columns contain ordered lists of Factors predicted to target the input list by the varous algorithms and associated libraries.
(TXT)

**S7 Table. CTCF interactors and rank-ordered algorithm outputs for Brain (WT_vs_CTCFko).** The first column contains proteins associated with CTCF extracted from the STRING database with a Combined Score>300. Subsequent columns contain ordered lists of Factors predicted to target the input list by the varous algorithms and associated libraries.
(TXT)

**S8 Table. FOS interactors and rank-ordered algorithm outputs for DGC(Quiet_vs_Reactive).** The first column contains proteins associated with FOS extracted from the STRING database with a Combined Score>300. Subsequent columns contain ordered lists of Factors predicted to target the input list by the varous algorithms and associated libraries.
(TXT)

**S9 Table. MAGIC Summary output for MCF7(shCon_vs_shREST).** The analysis statistics for the best scoring example of each Factor in the MAGIC matrix for MCF7(shCon_vs_shREST) are provided in the summary file.
(TXT)

**S10 Table. MAGIC Summary output for TCGA(Lum_vs_Basal).** The analysis statistics for the best scoring example of each Factor in the MAGIC matrix for TCGA(Lum_vs_Basal) are provided in the summary file.
(TXT)

**S11 Table. MAGIC Summary output for Brain(WT_vs_CTCFko).** The analysis statistics for the best scoring example of each Factor in the MAGIC matrix for Brain(WT_vs_CTCFko) are provided in the summary file.
(TXT)

**S12 Table. MAGIC Summary output for DGC(Quiet_vs_Reactive).** The analysis statistics for the best scoring example of each Factor in the MAGIC matrix for DGC(Quiet_vs_Reactive)

are provided in the summary file.
(TXT)

## Acknowledgments

I would like to thank Ray Dingledine and Steve Goldstein for critical insights into the statistics behind MAGIC. I thank John Svaren for helpful edits of the manuscript. I am grateful to Caroline Alexander for editing, aiding, encouraging and cojoling the submission of this manuscript. I would like to thank Queen because 'It's a Kind of Magic'.

## Author Contributions

**Conceptualization:** Avtar Roopra.

**Data curation:** Avtar Roopra.

**Formal analysis:** Avtar Roopra.

**Funding acquisition:** Avtar Roopra.

**Investigation:** Avtar Roopra.

**Methodology:** Avtar Roopra.

**Project administration:** Avtar Roopra.

**Resources:** Avtar Roopra.

**Software:** Avtar Roopra.

**Supervision:** Avtar Roopra.

**Validation:** Avtar Roopra.

**Visualization:** Avtar Roopra.

**Writing – original draft:** Avtar Roopra.

**Writing – review & editing:** Avtar Roopra.

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
