## [Decision Letter · Decision Letter 0]

1 Aug 2019

Dear Dr Roopra,

Thank you very much for submitting your manuscript 'MAGICTRICKS: A tool for predicting transcription factors and cofactors binding sites in gene sets using ENCODE data' for review by PLOS Computational Biology. Your manuscript has been fully evaluated by the PLOS Computational Biology editorial team and in this case also by independent peer reviewers. The reviewers appreciated the attention to an important problem, but raised some substantial concerns about the manuscript as it currently stands. While your manuscript cannot be accepted in its present form, we are willing to consider a revised version in which the issues raised by the reviewers have been adequately addressed. We cannot, of course, promise publication at that time.

Sincerely,

Ilya Ioshikhes

Associate Editor

PLOS Computational Biology

Thomas Lengauer

Methods Editor

PLOS Computational Biology

[LINK]

Reviewer's Responses to Questions

**Comments to the Authors:**

Reviewer #1: Minor:

1) Instead of ‘level’ use another term like ‘quantity’

2) I think part of your discussion about tissues heterogeneity should be put in the introduction.

Major (for me I think it is a Major revision but I will register this review only as minor revision because the validation of your algorithm can be done independently and later by other researchers)

1) Need validation of your algorithm: To be sure that your algorithm is interesting for life science research the validation of the prediction of the algorithm need to be done using 'wet lab' experiments. 'Cherry picking' online references is okay for 'discussion' but it not a validation of your algorithm.

Introduction

"We posit that in many cases, the majority of those changes are driven by alterations in the function of a few Factors that coordinate programmatic gene changes on a genome wide scale."

-> By opposition to clonal cell line, cancerous tissue is a mix of hundred different cell types. I think the observed change in gene transcript quantity is principally related to the change in cell quantity inside the sample. Please discuss this aspect of cell diversity in tissues analyzed and how this affects the observed changes in gene level quantity. e.g. : A disease tissues should have a drastic increase in the quantity of immune cells, Cancer tissues can show a significant change in the quantity of blood vessels...

Methods

-> Explain RNA extraction protocol. Trizol is an old technique, why not using Qiagen RNeasy columns?

"Probes were then collapsed to a single gene value ..."

-> 1 Gene with 1 value assigned to it is not what happen in reality. You need to talk about the problem of alternate splicing and multiple transcripts expressed per gene. Can the Transcription Factors have an influence on the complexity of RNA expression inside a single gene? Does the probes have been designed to target multiple transcripts per gene?

"any genes changed more than 3 Standard Deviations from the mean fold change..."

-> please explain more your methods to compute fold change of each gene. Which standard deviation are you using, is it the Standard Deviation between Probes from the same gene or between factors?

"Sams et al profiling..."

-> What is the meaning of ‘Sams et al profiling’ ?.

-> Explain RNAseq protocol

"were assigned a gene domain that was defined as 5Kb either side of the gene body.."

-> Explain why you choose 5Kb, why not 1KB? reference needed. Else can you add an option to your algorithm to scan for gene domain with a shorter or larger window and then report the most significant with some sort of ‘weight’ based on distance? (e.g.: More it is close of the RNA start and more it is significant)

" = -log (corr) × "

-> Can you please explain the scientific validity of this type of scoring (multiplication of transformed significance by ratio)? Do you assume that the log of significance (corrected pvalue) and the ratio of means will follow a similar distribution?, if not I think this scoring can induce a bias in the detection of more significance (lowest pValue) or to the opposite to the detection of bigger ratio, all this will dependent of the shape of the corresponding distribution of ‘Pcorr’ and ‘r’. Will it be possible to adapt your algorithm and add an option to rank the genes by ‘r’ using a fixed corr (e.g.: p< 0.01? or even p< 0.1?)

Results

-> Please define "FDR", is it the False Detecting Rate? how do you compute it? is it the same as corr? If it is False Detecting Rate can you use ‘%’, (e.g. FDR 5% instead of FDR 0.05)

-> ‘biologically meaningful’ can not be done by ‘cherry-picking’ online references or pathways because you will report only the reference that are validating your results and not the ones that go against. Your predicted results need to be validated by ‘wet lab’ experiments, what type of ‘wet lab’ experiments did you do to validate your informatics results?

Discussion

-> How does your algorithm react when more than 1 gene share the 5Kb window? Do you have some way to reduce the significance in function of the number of genes that share this TF (e.g.: if using a 5Kb windows you found 4 genes that share the same TF, only 1 gene is of interest I think the significance of this TF should be reduced because you don’t know which gene this TF is affecting)

-> I think you should put the discussion about tissues heterogeneity in the Introduction. Putting in the discussion make believe the reader that it is a limitation of your algorithm but it is in fact a limitation of the experiment and not of your algorithm. Every type of algorithm will be impaired if a list of gene from tissue with cell heterogeneity is used. The problem comes from the list used not from your algorithm. Gene expression from tissues with cell heterogeneity can be corrected by in-silico gene expression deconvolution prior to feeding to your algorithm (c.f. Dachet 2015, after analysis of correlation profile of gene expression they can be separated multiple list of genes could be fed to your algorithm, each list of genes represent 1 cell type).

Reviewer #2: The manuscript by Avtar Roopra presents a useful tool MAGICTRICKS which identifies enriched transcription factors (TFs) whose signals are enriched in genes (5'UTR-5kb to 3'UTR+5kb) in a query gene list (e.g. differentially expressed genes) compared to a master gene list (e.g., expressed genes). The tool identifies these enriched TFs using a 1-tailed Kolmogorov-Smirnov (KS) test of the max ChIP signals (from ENCODE) within genes (as detailed above) of the query gene list compared to the master gene list. A score is calculated using -log of the Benjamini-Hochberg corrected KS p-value x mean of the top 5th percentile query list ChIP values over mean of top (same number as query list) master list ChIP values. The resulting tool performs well against two other TF enrichment tools Enrichr and oPOSSUM-3 based on analysis of three datasets where TFs are knocked out or missing. However, there are a number of recently developed TF enrichment tools including BART, TFEA.ChIP, VIPER, DoRothEA v2 and ChEA3. In fact, in the recent ChEA3 manuscript (https://academic.oup.com/nar/article/47/W1/W212/5494769), the authors compare ChEA3 to these TF enrichment tools and MAGICTRICKS (which they call MAGICACT and reference the bioRxiv version of this submitted manuscript). They claim superior performance against all the TF enrichment tools including MAGICTRICKS. Given that the standard in the field is show some kind of advantage against existing top tools and to demonstrate this advantage by not just listing top genes (as is done in this manuscript) but compute performance metrics including precision, recall, AUC, etc (as is done in the ChEA3 manuscript), the following are the major and minor comments/suggested changes.

Major comments:

1. The author(s) should compare MAGICTRICKS to BART, TFEA.ChIP, VIPER, DoRothEA v2 and ChEA3 using performance metrics including precision, recall, AUC, etc and demonstrate some advantage. The perturbed TF datasets used in the ChEA3 manuscript allowed these calculations. These should be presented as main figures in the manuscript.

2. ChIP signals for each gene are defined by taking the max within the gene locus. Max is notoriously un-robust. A robust max (95th percentile of the ChIP signal within the gene locus) should also be implemented and available to users.

Minor comments:

1. This reviewer would not use a z-score cutoff of 1.65 for tools like oPossum. Something more like a z-score cutoff of 2 or higher should be used when comparing oPossum and MAGICTRICKS.

2. "Kolmogorov" is misspelled on p. 3 of the manuscript.

3. In Fig. 2 legend, "REST" should be replaced by "GATA3, ESR1 and FOXA1".

4. Under Fig. 2, a sentence mistakenly has "...yielded TFs with oPossum at z < 1.65..." which should be "...yielded TFs with oPossum at z > 1.65...".

5. Fig. 3A has "Down in" twice in the text highlighting the CTCF-KO down-regulated gene cluster in the heat map. In Fig. 3 legend, B) should be a "summary output for down-regulated genes" not "up-regulated" as written.

Reviewer #3: This manuscript describes MAGICTRICKS, a software tool for predicting factors responsible for the co-regulation of sets of genes. The tool relies on binding profiles from ChIP-seq experiments but, unlike other methods based on experimental data, it does not classify genes as targets and non-targets. Instead, each gene-factor pair is assigned a normalized ChIP-value and the program compares the distribution of values in the list of co-regulated genes and across all genes in the internal database. This is a major innovation that avoids the arbitrary classification of genes into two mutually exclusive categories.

Major comments:

1. As indicated by the author, a major limitation of the tool is that the current input matrix only considers peaks within 5Kb of the gene body. Firstly, aparently this distance was chosen arbitrarily. Second, and more important, many transcriptional regulators preferentially bind distant enhancers and thus most of theirs targets won’t be represented in the matrix. A potential solution to assign distant peaks to specific genes, is using the experimental evidences recorded in databases such as GeneHancer.

2. A second limitation is the relatively small set of Factors and Regulators included in MAGICTRICKS. The author could consider using ReMap instead of just data from the ENCODE project.

3. The selection of the highest ChIPseq value from a given factor across cell lines supposedly maximizes the likelihood that the site is universal, i.e. present in other cell types not represented in the original datasets. This hypothesis should be supported by references or data. In addition, this approach will blur tissue-specific or treatment-specific responses. Would not it be better to keep each experiment as an independent column in the internal matrix?

4. The examples used to illustrate the performance of MAGICTRICKS, although convincing, are just individual cases and thus do not provide compelling evidence for the general performance of the method. The use of complete collections of well-defined transcriptional signatures or gene sets, such as those in the GSEA MSigDB, will probably be more appropriate.

5. The Score for each transcription factor integrates the statistical significance of the KS test and a parameter (r) that tries to represents the size effect. Although we agree it is important to include both factors, the value of r is based on few extreme data points and, as such, it is not a indicators of the overall difference between master and query lists CDFs. We suggest the use of another parameters to represent the size effect (difference between both distributions) such as Ds-Di or the difference between the c value required to reach 0.5 of the CDF.

6. The tool is implemented for two specific operating systems. A cross-platform tool would be more widely accepted.

Minor comments:

1. The programming language used for the implementation is not stated.

2. Supplementary file 9 is not referenced in the manuscript.

3. There is a typo in the paragraph before the last in the Results section. It should say “supplemental file 12-Enrich tab” instead of “supplemental file 1-Enrich tab”

4. In the analysis of the CTCF knock-out experiment the conversion of mouse to human IDs is not explained.

5. The captions for figures 3B and 3C contradict the main text.

**Have all data underlying the figures and results presented in the manuscript been provided?**

Reviewer #1: Yes

Reviewer #2: Yes

Reviewer #3: Yes

PLOS authors have the option to publish the peer review history of their article (what does this mean?). If published, this will include your full peer review and any attached files.

Reviewer #1: Yes: Fabien DACHET

Reviewer #2: Yes: Stefan Bekiranov

Reviewer #3: Yes: Luis del Peso

---

## [Decision Letter · Decision Letter 1]

25 Feb 2020

Dear Dr. Roopra,

Thank you very much for submitting your manuscript "MAGIC: A tool for predicting transcription factors and cofactors driving gene sets using ENCODE data." for consideration at PLOS Computational Biology. As with all papers reviewed by the journal, your manuscript was reviewed by members of the editorial board and by several independent reviewers. The reviewers appreciated the attention to an important topic. Based on the reviews, we are likely to accept this manuscript for publication, providing that you modify the manuscript according to the review recommendations.

Sincerely,

Ilya Ioshikhes

Associate Editor

PLOS Computational Biology

Thomas Lengauer

Methods Editor

PLOS Computational Biology

[LINK]

Reviewer's Responses to Questions

**Comments to the Authors:**

Reviewer #1: The revised manuscript has been greatly improved, it is now clearer and addressed all my concerns. The new corrected algorithm accepts more parametrisations and I expect the authors will continue its development. I hope the authors will release updates that will allow to switch the TFs&factors analysis from genes to transcripts and splicing pattern identification using an adequate background file.

Reviewer #2: Dr. Roopra has addressed all my comments/suggestions in this revised manuscript.

There are just a few remaining minor issues that need to be addressed:

1. In addition to being made available from a University of Wisconsin website, MAGIC should be made available on GitHub, which serves as a more permanent site.

2. In the Methods section under the Precision Recall... subsection, "FET" should be defined as "Fisher's Exact Test" and there should be a period after "...each test (Calls Vectors)".

3. In the Methods section under the Rank Analysis subsection, D(r) is more clearly defined in the Results section as cumulative of fractional ranks for the manipulated Factor and PIs. I would use this definition in the methods section as well.

4. In the Methods section under the Rank Analysis subsection, the sentence "Blocks of positive calls in the Calls Vector were all assigned the rank of the first element of the block." is unclear. How are "blocks" defined? I was under the impression that every positive call was ranked.

5. In the Methods section under MCF7 RNA..., the last sentence is missing a period after "...(S1 Table)".

6. In the Methods section under Implementation of MAGIC in the second paragraph, "...each query list and generate a series..." should be "...each query list and generates a series...".

7. In the Results section in the Identification of Cofactors...subsection, "...either high or low ranks would produce a uniform distribution (which equates to r)" should be "...either high or low ranks would produce a uniform p-value distribution (which equates to r)" for clarity (unless something else was meant). it's known that a null model or random data with no enrichment will produce a uniform p-value distribution.

8. In the Discussion section, "...depending on the set of genes in it and so the maximal..." should be "...depending on the set of genes in it, and so the maximal...".

Best,

Stefan Bekiranov

Reviewer #3: The author have addressed most of our concerns and the current version of the method is clearly improved as compared to the original one. However, some concerns regarding the manuscript still remain, as detailed below.

Major comments:

1. The number of cases used to illustrate the performance of MAGIC have been slightly increased but, given the limited number of datasets and the arbitrary criteria for their selection, they still do not provide compelling evidence for the general performance of the method. At least the author should clearly acknowledge this important limitation when drawing conclusions. Particularly, when comparing MAGICTRICKS with the rest of tools included in the manuscript as reference.

2. The criteria to select genes of interest should be the same across the experiments used for validation. Even if the experiments weren't homogeneously process from reads to diferential expression. The author should provide a robust reason to not follow the same criteria when selecting genes for testing. Additionally, the author stated "Genes up-regulated more than 3 Standard Deviations from the mean fold change....". Since Fold Change doesn't follow a simmetric distribution, this apporach only serves to select up-regulated genes. Did the author meant the log-fold change?

3. The manuscript indicates that the tool is implemented for two specific operating systems, yet the address https://go.wisc.edu/MAGIC. only contains stand-alone for MacOS. More importantly, since the tool is written in python, the author could provide the source code so it can be used in any platform.

4. page 19 states that "plasticity[18]. 432 genes were down-regulated upon CTCF loss out of 17,015 expressed genes (S3 Table).". However, according to the results reported on GEO, there are 321 genes with log2(FC)<0 and adjusted p-value <0.05. If the author has analized the RNA-seq following a different pipeline that the one reported on GEO, it should be stated.

5. On page 23 the author states "Nevertheless, it provided an extremely powerful prediction of EZH2 induction which was missed by CHEA3 and TFEA but was confirmed

by Enrichr.". However, we analyzed the dataset provided as supplementary information in reference 43 (https://journals.plos.org/plosone/article?id=10.1371/journal.pone.0226733#sec033) and found that TFEA identifies EZH2 (along with SUZ12 and JARID2) as the top candidate for this dataset. Moreover, five of the top ten datasets represent EZH2 ChIPseqs. In the case of ChEA3 (literature library), the top candidate is MTF2 (also known as Polycomb-Like 2) that mediates binding of PRC2 to DNA. The author should correct their conclusion regarding the performance of TFEA and ChEA3 or provide a compelling argument to explain the indicated discrepancies.

Minor comments:

1. It's possible to ask ReMap's authors for the MACS2 outputs of their ChIP collection. The current ENCODE dump has a wide range of TF covered, but you might want to consider ReMap for a future update.

2. Methods. Page 15. "An emperical distribution function...", should say "empirical".

3. Figure legends page 24, spelling of "reciprical"

3. Last line pag 16 says "ChIP value at dsup", for consistency "dsup" should by capitalized as "Dsup"

4. figure 1, please indicate the meaning of "ND".

5. Quality of figures 2 and 3 is poor.

**Have all data underlying the figures and results presented in the manuscript been provided?**

Reviewer #1: Yes

Reviewer #2: Yes

Reviewer #3: None

PLOS authors have the option to publish the peer review history of their article (what does this mean?). If published, this will include your full peer review and any attached files.

Reviewer #1: Yes: Fabien DACHET

Reviewer #2: Yes: Stefan Bekiranov

Reviewer #3: No
---

## [Decision Letter · Decision Letter 2]

19 Mar 2020

Dear Dr. Roopra,

We are pleased to inform you that your manuscript 'MAGIC: A tool for predicting transcription factors and cofactors driving gene sets using ENCODE data.' has been provisionally accepted for publication in PLOS Computational Biology.

Please add on that occasion a link in the manuscript to the GitHub page where the Python script and information on how to access the other relevant files are deposited, as per the Reviewer 2 request.

Best regards,

Ilya Ioshikhes

Associate Editor

PLOS Computational Biology

Thomas Lengauer

Methods Editor

PLOS Computational Biology

Reviewer's Responses to Questions

**Comments to the Authors:**

Reviewer #2: Dr. Roopra has addressed all my remaining minor comments/suggestions.

The only thing that remains is adding a link in the manuscript to the GitHub page where the Python script and information on how to access the other relevant files are deposited.

Best,

Stefan Bekiranov

Reviewer #3: The author have has addressed all our comments/suggestions in the R2 version of the manuscript.

**Have all data underlying the figures and results presented in the manuscript been provided?**

Reviewer #2: Yes

Reviewer #3: Yes

PLOS authors have the option to publish the peer review history of their article (what does this mean?). If published, this will include your full peer review and any attached files.

Reviewer #2: Yes: Stefan Bekiranov

Reviewer #3: No

---

## [Editor Report · Acceptance letter]

30 Mar 2020

PCOMPBIOL-D-19-00993R2 

MAGIC: A tool for predicting transcription factors and cofactors driving gene sets using ENCODE data.

Dear Dr Roopra,

I am pleased to inform you that your manuscript has been formally accepted for publication in PLOS Computational Biology. Your manuscript is now with our production department and you will be notified of the publication date in due course.

With kind regards,

Laura Mallard
